# Epidemiology and Antimicrobial Resistance of Uropathogens in a Tertiary Care Center in Riyadh, Saudi Arabia: A One-Year Retrospective Analysis

**DOI:** 10.3390/pathogens14121254

**Published:** 2025-12-08

**Authors:** Fizza Khalid, Wael Jumah Aljohani, Adeel Akram, Abdullah Bukhari, Osamah T. Khojah

**Affiliations:** 1Department of Microbiology, MDLab Dr. Sulaiman Al Habib Medical Group, Riyadh 11325, Saudi Arabia; 2Research Center, Dr. Sulaiman Al Habib Medical Group, Riyadh 14255, Saudi Arabia; 3Department of Molecular and Genetics, MDLab Dr. Sulaiman Al Habib Medical Group, Riyadh 11325, Saudi Arabia; 4Department of Medicine, Infectious Diseases, Dr. Sulaiman Al Habib Medical Group, Riyadh 14212, Saudi Arabia; 5Internal Medicine Department, College of Medicine, Imam Mohammad Ibn Saud Islamic University, Riyadh 11564, Saudi Arabia; 6Department of Pathology, College of Medicine, King Saud University, Riyadh 11472, Saudi Arabia; 7Laboratory Department, Chief Medical Officer Office, Dr. Sulaiman Al Habib Medical Group, Riyadh 11643, Saudi Arabia

**Keywords:** extended spectrum beta lactamase (ESBL), uropathogens, urinary tract infection (UTI), antimicrobial resistance

## Abstract

Urinary tract infections (UTIs) represent one of the most frequent bacterial infections worldwide, with *Escherichia coli* (*E. coli*) and *Klebsiella pneumoniae* (*K. pneumoniae*) as the predominant uropathogens. The emergence of extended-spectrum β-lactamase (ESBL)-producing Enterobacterales has severely limited treatment options, making regional surveillance crucial. This study aimed to determine the prevalence of uropathogens, assess antimicrobial resistance patterns, and evaluate the burden of ESBL-producing organisms among patients presenting with suspected UTIs in a tertiary care hospital in Riyadh. We conducted a retrospective analysis of 19,556 urine cultures from a tertiary care hospital in Riyadh, Saudi Arabia, between January and December 2024. Of these, 2629 (13.4%) cultures showed significant bacterial growth, predominantly in females (83.2%) and in the 16–30 year age group. *E. coli* accounted for 65.9% of isolates, followed by *K. pneumoniae* (16.8%). ESBL production was detected in 28.5% of *E. coli* and *Klebsiella* isolates. ESBL producers exhibited complete resistance to third-generation cephalosporins and β-lactam/β-lactamase inhibitor combinations, whereas carbapenems, aminoglycosides, and fosfomycin maintained high efficacy. Resistance to ciprofloxacin and co-trimoxazole was widespread in both ESBL and non-ESBL isolates. Additionally, vancomycin-resistant enterococci (7%), methicillin-resistant *Staphylococcus aureus* (2%), and carbapenem-resistant Enterobacterales (0.9%) were found. These findings highlight the escalating burden of ESBL-associated UTIs and underscore the urgent need for strengthened antimicrobial stewardship, continuous surveillance, and optimized empirical therapy to mitigate the impact of multidrug-resistant uropathogens in clinical practice.

## 1. Background

UTIs are among the most common bacterial infections worldwide, affecting millions annually and posing significant healthcare challenges [1]. They are far more prevalent in women due to anatomical and physiological susceptibility [2,3]. UTIs range from mild cystitis to severe complications such as pyelonephritis and urosepsis, particularly in high-risk groups including the elderly, immunocompromised patients, and individuals with comorbidities [4]. *Escherichia coli* (*E. coli*) remains the predominant uropathogen globally, responsible for 60–80% of community-acquired infections, while *Klebsiella pneumoniae* (*K. pneumoniae*), *Enterococcus faecalis*, and *Pseudomonas aeruginosa* also contribute significantly, particularly in nosocomial or recurrent infections [5]. Epidemiological studies conducted in the Middle East and Saudi Arabia confirm similar trends, with *E. coli* as the leading causative agent, followed by *Klebsiella* and other Enterobacterales species, though notable geographical variability has been reported [6,7]. The burden of UTIs in Saudi Arabia is compounded by high outpatient presentation rates, which pose challenges for empirical treatment selection due to evolving resistance patterns. Recent regional surveillance reports indicate considerable variability in antimicrobial resistance across different provinces in Saudi Arabia, with ESBL prevalence and susceptibility profiles differing between healthcare settings and populations. However, there remains a lack of uniform, up-to-date data consolidating these variations at the institutional level, highlighting a critical gap in guiding empirical therapy and stewardship strategies.

The global increase in antimicrobial resistance (AMR) among uropathogens has become a major public health concern, with ESBL-producing Enterobacterales at the forefront of this crisis. ESBL enzymes confer resistance to most β-lactams, including third-generation cephalosporins, and are frequently associated with resistance to other drug classes such as fluoroquinolones and trimethoprim-sulfamethoxazole, limiting treatment options [8]. The prevalence of ESBL-producing organisms varies by region, with recent studies from Saudi Arabia reporting rates ranging from 15% to over 40% among Enterobacterales isolates [9,10]. Comparable prevalence rates have been observed in neighboring countries such as Iran and Pakistan, highlighting the widespread dissemination of ESBL phenotypes across the Middle East and South Asia [11,12].

Risk factors for ESBL colonization and infection include prior exposure to broad-spectrum antibiotics, prolonged hospital stays, intensive care admission, urinary catheterization, diabetes, and recurrent UTIs [13]. The inappropriate use of cephalosporins and fluoroquinolones in particular has been strongly associated with the emergence of resistant clones, which often carry ESBL genes on mobile genetic elements co-harboring other resistance determinants [14]. This genetic linkage explains the high co-resistance observed to fluoroquinolones and co-trimoxazole among ESBL producers [15].

Carbapenems remain the most reliable therapeutic option for ESBL associated infections; however, their widespread use has contributed to the emergence of carbapenem-resistant Enterobacterales (CRE), further complicating treatment. Alternative agents such as aminoglycosides, fosfomycin, and nitrofurantoin continue to demonstrate good activity against many ESBL producers and are increasingly considered in both inpatient and outpatient management [16]. Newer β-lactam/β-lactamase inhibitor combinations, such as ceftazidime–avibactam, provide additional therapeutic options but remain limited by cost and accessibility in many regions [17].

The threat of multidrug-resistant organisms (MDROs) extends beyond ESBL producers. Vancomycin-resistant *Enterococci* (VRE), methicillin-resistant *Staphylococcus aureus* (MRSA), and CRE have increasingly been reported as uropathogens, particularly in healthcare-associated settings. VRE is associated with prolonged hospitalization and catheterization, while MRSA urinary infections are more commonly observed in community settings and may arise from hematogenous spread [18,19]. These pathogens significantly limit therapeutic options and are associated with worse clinical outcomes.

Given the high prevalence of resistant uropathogens, particularly ESBL-producing Enterobacterales, and the rising threat of VRE, MRSA, and CRE, continuous local surveillance is essential for guiding empirical therapy. Antimicrobial stewardship programs, tailored to regional resistance patterns, are urgently needed to optimize treatment, prevent therapeutic failures, and slow the further emergence of resistance. This underscores the importance of studies evaluating both the epidemiology of uropathogens and their resistance profiles within specific geographic and clinical contexts, such as the present investigation from Saudi Arabia. Accordingly, this study aims to (1) assess the prevalence and resistance patterns of uropathogens isolated from UTIs in our settings and (2) determine the burden of ESBL-producing and other multidrug-resistant phenotypes to support evidence-based empirical treatment and antimicrobial stewardship efforts.

## 2. Methodology

### 2.1. Demographics of Study Participants and Setting

This retrospective study included 19,556 urine samples submitted for culture from both inpatients and outpatients between January and December 2024. Urine culture results with bacterial counts ≥10^4^ CFU/mL were considered significant in cases where the sampling technique, patient characteristics, or clinical presentation suggested a high likelihood of true infection. The conventional threshold of 10^5^ CFU/mL remains the standard criterion for clean-catch midstream specimens; however, due to incomplete clinical information and absence of consistent physician documentation in our dataset, adopting the lower threshold constitutes a methodological limitation of the study.

For samples yielding colony counts below 10^5^ CFU/mL, the microbiology laboratory routinely contacts the treating physician to obtain relevant clinical information (symptoms, pregnancy status, catheterization, previous antibiotic use) before reporting the significance of the isolate. This step was applied operationally but not uniformly captured in the electronic data; hence it was added here for methodological clarity.

Urine cultures positive for fungal isolates were excluded. The study was performed at a 450-bed tertiary care hospital in Riyadh, Saudi Arabia. Demographic variables, including age and sex, were retrieved from the electronic medical records. Samples with incomplete demographic or clinical details were excluded, as illustrated in Figure 1 (STROBE flow diagram).

### 2.2. Bacterial Identification and Antimicrobial Susceptibility Testing

All urine specimens were collected as midstream clean-catch samples by patients and transported to the microbiology laboratory. The laboratory operates 24 h a day, allowing specimens to be received and processed promptly. Under standard operating procedures, urine samples were inoculated within one hour of arrival to preserve colony counts and prevent bacterial overgrowth.

A calibrated 1 µL disposable loop was used to inoculate each specimen onto Blood Agar (Oxoid, Basingstoke, UK) and CLED Agar (Oxoid). Plates were incubated at 35–37 °C for 24–48 h. The extended 48-hour incubation was used in cases where mixed growth patterns required further clarification, slow-growing Gram-positive organisms were suspected, or low colony counts required additional incubation for confirmatory growth.

Significant bacterial growth was identified using the Vitek 2 Compact System (bioMérieux, Marcy I’Etoile, France) following the manufacturer’s instructions. Minimum inhibitory concentrations (MICs) were interpreted according to CLSI 2024 guidelines.

Although the Vitek 2 system was used as the primary method for identification and susceptibility testing, confirmatory were performed when results were questionable or resistance mechanisms were suspected. These included double-disc synergy test (DDST) for ESBL confirmation and disk diffusion (Oxoid) when Vitek results showed inconsistencies or uncommon resistance patterns. All secondary tests were performed according to CLSI recommendations using standard media (Mueller-Hinton agar; Oxoid, Basingstoke, UK).

### 2.3. Ethical Approval and Statistical Analysis

The institutional review board of Dr Sulaiman Al Habib Medical Group approved this study (RC25.03.37). Statistical analysis was conducted using SPSS (IBM Co., Armonk, NY, USA). Categorical data were expressed in terms of frequency and percentage and analyzed using the chi-square test or exact test, as applicable. A *p*-value < 0.05 was considered statistically significant for all comparative analyses.

## 3. Results

During the study period, 19,556 urine cultures were submitted from inpatients and outpatients; 2629 (13.4%) yielded significant bacterial growth and were included in the analysis. Table 1 summarizes the demographic distribution and species-level breakdown of positive cultures. Overall, *E. coli* was the dominant uropathogen with 1732/2629 (65.9%) of positive cultures (Table 1), and it was predominantly isolated from females, particularly those aged 16–30 years and 31–45 years. The organism distribution across age groups showed similar proportions for major pathogens such as E. coli, reflecting a uniform burden across age categories rather than age-specific clustering. *K. pneumoniae* was the second most common isolate (442/2629; 16.8%) and showed a similar age and sex distribution to *E. coli*. Enterococci accounted for 129/2629 (4.9%) of isolates overall, with *E. faecalis* being the most frequent enterococcal species and occurring more commonly in older age groups (61–75 and >75 years). The total number of *S. aureus* isolates was 19 (0.7%); other coagulase-negative staphylococci (CoNS) were uncommon and are discussed below.

Figure 2 shows the distribution of test results by gender. The majority of tested females were culture-negative (58.7%, n = 11,487). Among positive cultures, females represented 83.3% of positive isolates (n = 2188) and males 16.7% (n = 441). The largest number of positive cultures occurred in the 31–45 year age group (n = 834), followed by 16–30 years (n = 398) and 46–60 years (n = 353).

We found 28.5% ESBL producers (*E. coli:534*, *Klebsiella* species:109). Table 2 compares antibiotic resistance patterns between non-ESBL and ESBL-producing *E. coli* isolates. ESBL producers showed significantly higher resistance to most beta-lactam antibiotics and to fluoroquinolones (all *p* < 0.001 for key comparisons). Aminoglycosides (amikacin, gentamicin) and carbapenems maintained low resistance rates across both groups. Table 3 presents analogous comparisons for *Klebsiella* species.

Figure 3 displays the susceptibility profile of enterococci: linezolid retained full activity against all tested isolates (100% susceptible), while vancomycin resistance was detected in 7% of enterococcal isolates. Ampicillin and nitrofurantoin showed high activity (85.3% and 82.9% susceptible, respectively), whereas tetracycline susceptibility was low (34.9%).

Figure 4 shows *Staphylococcus* species antibiograms. Oxacillin susceptibility was 78.9% among tested *S. aureus* isolates; gentamicin, vancomycin and linezolid showed high activity. The number of CoNS isolated from urine was small; given the clinical context and growth characteristics, many CoNS isolates were considered likely contaminants

Figure 5 summarizes *P. aeruginosa* susceptibility: aminoglycosides (amikacin 97.2%, gentamicin 98.1%) and carbapenems (meropenem 94.4%, imipenem 91.7%) demonstrated high activity; ceftazidime and piperacillin/tazobactam also remained largely effective (>90% susceptible). A majority of both total cultures and ESBL-positive isolates originate from outpatients, with 91.7% of ESBL-positive cases detected in this group, compared to only 8.3% from inpatients.

The dendrograms visualize how antibiotics cluster based on similarities in susceptibility patterns. These dendrograms, shown in Figure 6 and Figure 7, depict the grouping of antibiotic resistance profiles, presumably contrasting ESBL-producing organisms with non-ESBL-producing organisms, according to susceptibility patterns to the tested antibiotics. Antibiotics placed close together exhibit similar resistance behavior across isolates. In ESBL-producers, cephalosporins and fluoroquinolones form a tight cluster, reflecting shared resistance mechanism. In contrast, carbapenems and aminoglycosides cluster separately, indicating preserved activity. Non-ESBL isolates show more dispersed clustering, representing broader susceptibility.

Figure 6 illustrates the hierarchical clustering of antibiotic susceptibility profiles for ESBL-producing microbes. Two primary clusters are observable: One encompasses broad-spectrum antibiotics such as carbapenems (imipenem, meropenem, and ertapenem), fosfomycin, aminoglycosides (amikacin and gentamicin), and piperacillin/tazobactam. These are generally more efficacious against ESBL organisms. The alternative cluster comprises cephalosporins (such as ceftriaxone and cefpodoxime) and co-trimoxazole, ciprofloxacin, which have diminished efficacy against ESBL production due to resistance mechanisms.

Figure 7 illustrates the clumping of non-ESBL-producing species. The clustering exhibits increased dispersion, signifying heightened antibiotic susceptibility. Antibiotics such as ceftriaxone, cefpodoxime, and co-trimoxazole are less closely associated with antibiotic resistance, indicating that non-ESBL organisms exhibit greater susceptibility to a wider array of antibiotics.

The distribution of VRE, MRSA, and CRE cases across inpatient and outpatient settings demonstrated distinct epidemiological patterns as shown in Table 4. For VRE, the majority of negative results originated from outpatients, whereas positive cases were predominantly detected among inpatients, reflecting a higher inpatient-associated burden. MRSA showed a different profile: both negative and positive cases were more frequently reported among outpatients, although the small number of positive cases warrants cautious interpretation. In the case of CRE, most negative specimens were contributed by outpatients due to the larger outpatient sample volume; however, the proportion of CRE-positive isolates was considerably higher among inpatients, consistent with the typical hospital-acquired nature of CRE. Overall, VRE and CRE positivity showed stronger inpatient associations, while MRSA positivity appeared more evenly distributed but numerically higher in the outpatient group.

## 4. Discussion

UTIs, which can particularly afflict women of all ages, are the most common bacterial infections in clinical practice. A significant rise in the global incidence of UTIs has been observed in past two decades in both men and women; however, the incidence is four times higher in women as compared to men [20]. Depending on comorbidities and risk factors that predispose the patient, it might vary from cystitis to urosepsis and septic shock. Timely detection of uropathogen with susceptibility and antibiotic resistance pattern detection is vital to guide appropriate therapy, reduce complications, prevent resistance, and improve patient outcome. This retrospective study analyzed 19,556 urine culture samples collected over one year from both inpatients and outpatients at a tertiary care hospital in Riyadh, Saudi Arabia. Of these, 13.4% yielded significant bacterial growth, highlighting the substantial burden of UTIs in the population studied. Our results are slightly lower than other studies published from Saudi Arabia. Almutawif, and Eid reported 23.4% urine positive for the significant bacterial growth [21]. Similarly, another study from Hail city also reported 19.6% of positive urine cultures among all the tested urine samples [22]. However, the prevalence of UTI has been reported to range from 10% to as high as 32% in different countries [23,24,25]. These disparities could be because of geographic distribution, sample size variation, and hospital practice and protocol followed.

Females accounted for a higher proportion of positive cultures (83.2%), with the majority of cases occurring in the 16–30 and 31–45 age groups. These findings are consistent with previous studies, which have shown a higher prevalence of UTIs among females due to anatomical and physiological factors, including a shorter urethra and hormonal influences [21,25,26]. The greater incidence rate of UTI in females ranging from 30 to 49 years is also evident in a recent epidemiological report by He et al. [20]. The highest occurrence of uncomplicated UTI occurs during the peak years of sexual activity, which are typically in parallel with our findings; however, the incidence of UTI increases with age depending on many factors like menopause and co-morbidities [27].

*E. coli* emerged as the most frequently isolated uropathogen, constituting 65.9% of all positive cultures, followed by *K. pneumoniae* (16.8%). This aligns with global epidemiological trends, where *E. coli* remains the predominant causative agent of community and hospital-acquired UTIs. Notably, *E. coli* was especially prevalent in younger females, reflecting known patterns of uncomplicated UTIs in this demographic [28]. Other significant pathogens included *Enterococcus faecalis*, *Pseudomonas aeruginosa*, and *Staphylococcus saprophyticus*, each demonstrating specific age and gender associations.

Antimicrobial resistance is a worldwide concern. Alterations in bacterial genes and the emergence of antibiotic resistance are mostly attributable to the inappropriate utilization of broad-spectrum antibiotics, excessive prescribing by healthcare professionals, and poor usage by patients [29]. Bacteria that make ESBL are becoming more common in UTIs, and there are a number of reasons why this is happening. One of the most important factors is having used broad-spectrum antibiotics before, especially cephalosporins and fluoroquinolones, which make bacteria resistant by putting selection pressure on them [30]. Hospitalizations, stays in the ICU, and urinary catheters are all frequent dangers that come with getting medical care, especially when infection control is insufficient. Factors connected to the patient, such as repeated UTIs, immunosuppression, age, and comorbidities like diabetes, make them even more vulnerable [31]. These dangers together show how important it is to take antibiotics carefully and to utilize focused preventative methods in people who are at high risk. ESBL prevalence is even higher in Pakistan with reporting rates exceeding 40% [32]. In contrast, broader regional surveillance data from Gulf Cooperation Council (GCC) countries report ESBL-producing Enterobacterales prevalence ranging from approximately 21.6% to 29.3% across general clinical isolates, reflecting a substantial but variable burden within the region [33]. These differences highlight the substantial regional variation in AMR epidemiology and underscore the need for country-specific surveillance systems.

Local research in Saudi Arabia has identified elevated levels of antibiotic resistance and ESBL bacteria in multiple regions [29,34,35]. We found 28.5% (627/2200) of *E. coli* and *Klebsiella* species as ESBL producers. Many studies have reported a similar trend of ESBL from clinical isolates of Enterobacterales [36,37]. This finding is also supported by Abalkhail et al., who revealed a considerable prevalence of ESBL-producing organisms among Enterobacterales [38]. Another study from Iran also showed 37.11% strains of *E. coli* as ESBL producers [39]. However, our percent isolation of ESBL is comparatively higher than a study reported from AL Baha region as 15% [9]. These organisms showed high resistance to third-generation cephalosporins and β-lactam/β-lactamase inhibitor combinations, posing significant therapeutic challenges. In contrast, carbapenems (imipenem, meropenem, and ertapenem), aminoglycosides (amikacin and gentamicin), and fosfomycin retained high activity against both ESBL and non-ESBL-producing isolates. These findings reinforce the importance of preserving the use of carbapenems for severe infections and highlight fosfomycin as a potential oral agent against resistant strains, particularly in outpatient settings.

A comparative analysis of antibiotic susceptibility showed significant differences between ESBL and non-ESBL-producing organisms. Most cephalosporins were ineffective against ESBL producers, as indicated by resistance in all tested isolates. The 81.6% ciprofloxacin resistance in ESBL-producing *E. coli* has clear clinical implications, ciprofloxacin should not be used empirically for UTI treatment in this population. Conversely, the high susceptibility to fosfomycin (97.2%) and nitrofurantoin (94.5%) supports their continued use as effective oral agents for uncomplicated UTIs. Ciprofloxacin and co-trimoxazole also exhibited high resistance rates in both groups, raising concerns about their continued empirical use in UTI treatment. Quinolones showed a significant relation with ESBL production, as 81.6% ciprofloxacin resistance was found in ESBL-producing *E. coli*. These findings are supported by Azargun et al. who also found quinolone resistance rate of more than 80% of ESBL producers [40]. One possible explanation for the co-occurrence of resistance to β-lactams and fluoroquinolones is that the existence of ESBL and some fluoroquinolone-resistant genes within the same mobile genetic components could be the cause. Conversely, amikacin and carbapenems demonstrated consistent efficacy across both bacterial groups, suggesting their utility in empiric treatment of complicated UTIs or in settings with high ESBL prevalence.

The high resistance rates to ciprofloxacin and co-trimoxazole, two agents historically used as first-line therapy, have major clinical implications. Their continued empirical use in the region risks therapeutic failure, prolonged illness, and increased healthcare utilization. These findings reinforce the need to revise local prescribing practices and discourage the empirical use of these agents for uncomplicated UTIs unless susceptibility results are available.

Hierarchical clustering of antibiotic susceptibility patterns via dendrograms further emphasized the divergence in response between ESBL and non-ESBL-producing bacteria. For ESBL producers, carbapenems and aminoglycosides formed distinct clusters indicative of sustained efficacy, while cephalosporins, fluoroquinolones, and co-trimoxazole were grouped as less effective. Non-ESBL strains displayed broader susceptibility profiles, with fewer tightly clustered resistance patterns, highlighting a greater therapeutic window for these organisms.

The prompt and accurate identification of ESBL-associated UTIs continues to be a major difficulty. Conventional culture-based methods, while widely used, are slow, and many laboratories do not have access to specific ESBL detection assays. Although molecular diagnostics provide faster and more reliable results, they are not consistently available in daily practice, leading to delays in therapy and reliance on empirical regimens that may be inappropriate [41]. Management is further complicated by the extensive resistance patterns of ESBL producers. Carbapenems are frequently considered the most dependable option for severe infections; however, their widespread use risks accelerating carbapenem resistance [42]. For uncomplicated infections, oral alternatives such as fosfomycin and nitrofurantoin remain useful, while newer β-lactam/β-lactamase inhibitor combinations offer additional therapeutic potential but remain costly and less accessible in many healthcare settings [16].

The detection of multidrug-resistant organisms (MDROs) such as vancomycin-resistant *Enterococcus* (VRE), MRSA, and carbapenem-resistant Enterobacterales (CRE) among uropathogens in this study highlights a growing concern for urinary tract infection (UTI) management. The predominance of VRE in inpatients (77.8%) suggests that nosocomial factors, such as prolonged catheterization and prior antibiotic exposure, may drive colonization and infection risk. VRE has increasingly been recognized as an emerging uropathogen, complicating treatment due to limited therapeutic options [43]. Similarly, the presence of CRE (40%) is worrisome given the reliance on carbapenems for treating complicated UTIs caused by multidrug-resistant Gram-negative bacilli. CRE uropathogens are associated with high morbidity and mortality, particularly in hospitalized or immunocompromised patients [41].The presence of CRE in 40% of isolates, especially the high outpatient proportion, indicates a worrisome shift toward community-associated resistance.

In contrast, MRSA was more frequently detected in outpatients (75%), suggesting that community-associated strains may play a role in urinary infections. While *S. aureus* is an uncommon cause of community-acquired UTI, its detection in outpatients raises the possibility of hematogenous spread or ascending infection, which has been increasingly described in the literature [18]. The outpatient predominance of MRSA may also reflect community-associated transmission dynamics, necessitating surveillance beyond hospital settings.

Preventing infections caused by ESBL-producing organisms and other MDROs requires strong infection control practices and effective antimicrobial stewardship. Core strategies include meticulous hand hygiene, minimizing catheterization, and maintaining strict environmental cleaning protocols. Antimicrobial stewardship programs are essential to ensure rational prescribing, reduce inappropriate antibiotic use, and optimize patient outcomes. These programs, supported by local antibiograms and continuous surveillance, enable evidence-based treatment decisions. Furthermore, raising awareness among healthcare workers and the public about prudent antibiotic use is critical to slow the emergence of resistance. Collectively, these approaches represent the foundation of strategies to limit the impact of ESBL infections in hospitals as well as in community settings. These findings suggest that current empirical therapy guidelines for UTIs require re-evaluation. Nitrofurantoin and fosfomycin remain reliable first-line options for uncomplicated infections, while carbapenems should be preserved for complicated or ESBL-associated UTIs. Future regional clinical guidelines must consider emerging community-associated CRE and MRSA patterns to avoid under-treatment and adverse outcomes.

Despite the large sample size and use of standardized laboratory methods, the study has several limitations. Its retrospective, single-center design may limit generalizability to other institutions or regions. The absence of clinical data restricted our ability to perform risk factor analysis or correlate microbiological findings with clinical outcomes. Potential selection bias related to culture submission practices cannot be excluded. Additionally, molecular characterization of resistance genes was not performed, which could have provided deeper insights into the genetic mechanisms underlying ESBL and CRE production. A major limitation of this study is the absence of patient-level clinical data, which prevented analysis of risk factors, comorbidities, catheter use, symptom severity, and treatment outcomes. As a result, microbiological findings could not be correlated with clinical presentations, therapeutic responses, or recurrence rates.

Given the high rates of resistance to commonly used antibiotics such as ciprofloxacin and co-trimoxazole, empirical treatment protocols should be re-evaluated regularly based on local susceptibility data. Enhanced antimicrobial stewardship programs, particularly in outpatient settings where resistance is rising, are crucial. Regular surveillance and molecular typing of resistant isolates are recommended to monitor trends and guide infection control strategies. Finally, future studies should adopt multicenter, prospective designs and include patient-level clinical data for more comprehensive risk assessments. Given these resistance trends, antimicrobial stewardship programs in Saudi Arabia should prioritize restricting unnecessary fluoroquinolones and broad-spectrum beta lactam use, implementing indication based order sets, and promoting narrow-spectrum options guided by local antibiograms. Regular prescriber education, audit and feedback strategies, and integration of rapid diagnostics into routine practice can further enhance stewardship effectiveness.

In conclusion, this study highlights the continued dominance of *E. coli* and *Klebsiella* species in UTIs, the alarming prevalence of ESBL-producers, and the emerging community circulation of CRE and MRSA. While carbapenems, aminoglycosides, and fosfomycin remain effective against resistant organisms, the narrowing of empirical treatment options emphasizes the urgent need for targeted antimicrobial use, strengthened outpatient stewardship, and robust infection prevention efforts.

Future research should involve prospective multicenter studies incorporating detailed clinical variables and molecular characterization of resistance genes to better define risk factors, transmission dynamics, and evolving resistance landscape in Saudi Arabia.

## Figures and Tables

**Figure 1 pathogens-14-01254-f001:**
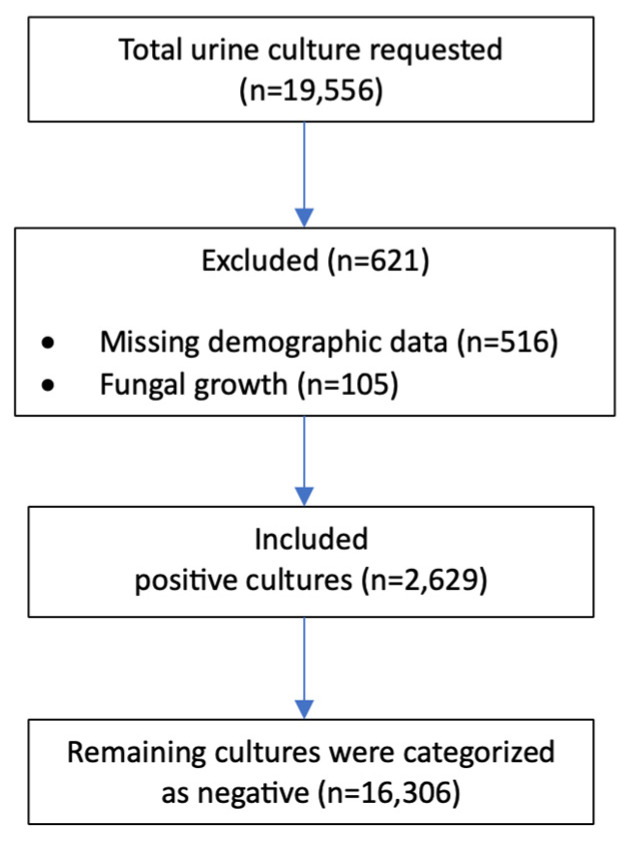
STROBE flow chart.

**Figure 2 pathogens-14-01254-f002:**
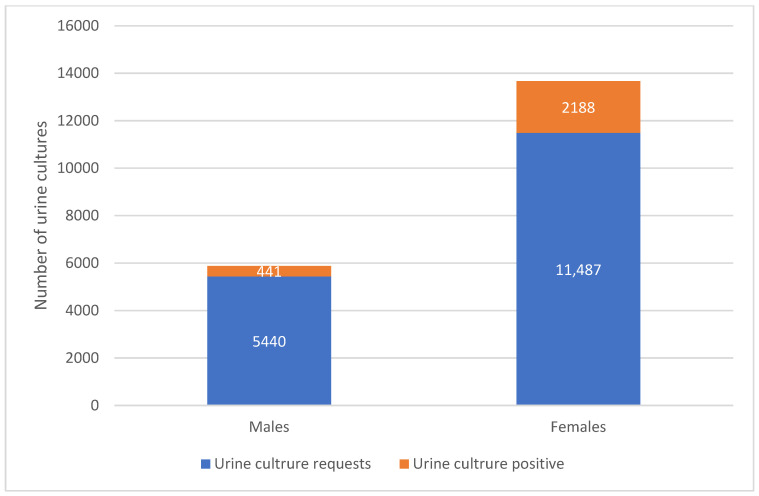
Distribution of Urinary Tract Infection Cases by Gender and Test Results. The figure compares the total number of urine culture requests and culture-positive cases among males and females.

**Figure 3 pathogens-14-01254-f003:**
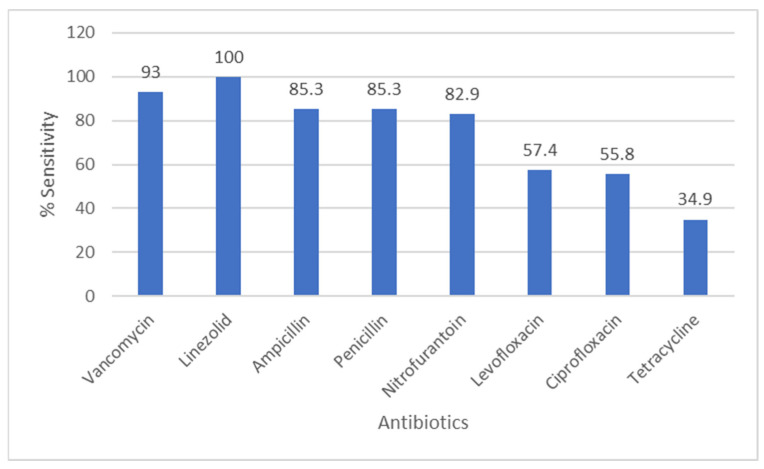
Antibiotic Sensitivity Rates of *Enterococcus* Species. The figure shows antibiotic sensitivity patterns of *Enterococcus* species, showing the proportion of isolates susceptible to commonly tested antimicrobial agents.

**Figure 4 pathogens-14-01254-f004:**
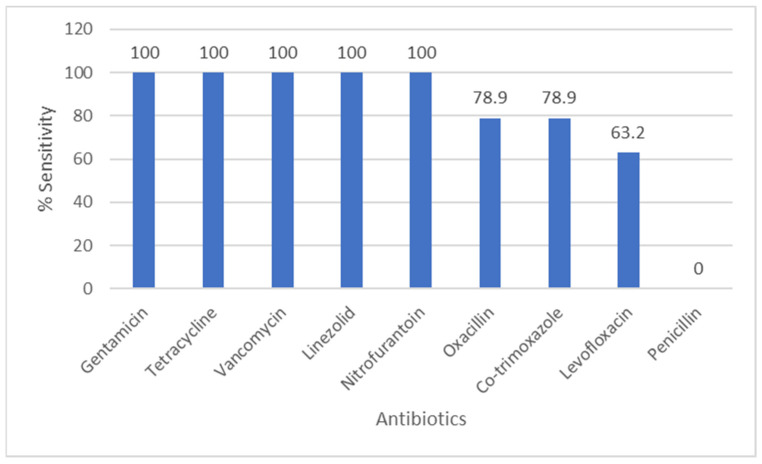
Antibiotic Sensitivity Rates of *Staphylococcus* Species. The figure shows antibiotic sensitivity patterns of *Staphylococcus* species, showing the proportion of isolates susceptible to commonly tested antimicrobial agents.

**Figure 5 pathogens-14-01254-f005:**
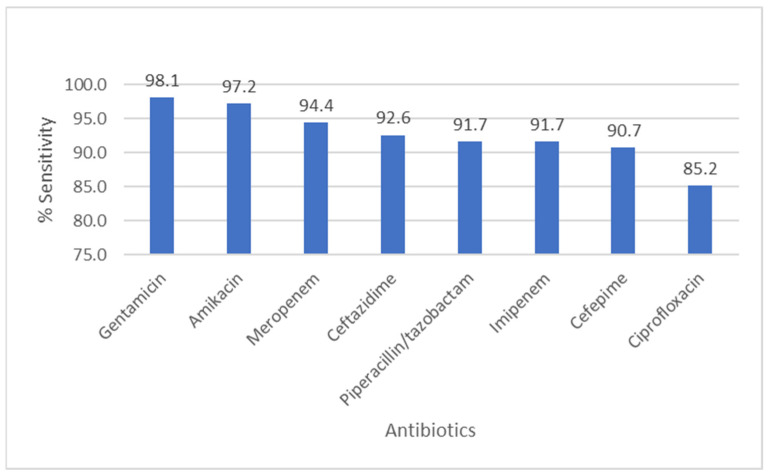
Antibiotic Sensitivity Rates of *Pseudomonas aeruginosa.* The figure shows antibiotic sensitivity patterns of *Pseudomonas aeruginosa*, showing the proportion of isolates susceptible to commonly tested antimicrobial agents.

**Figure 6 pathogens-14-01254-f006:**
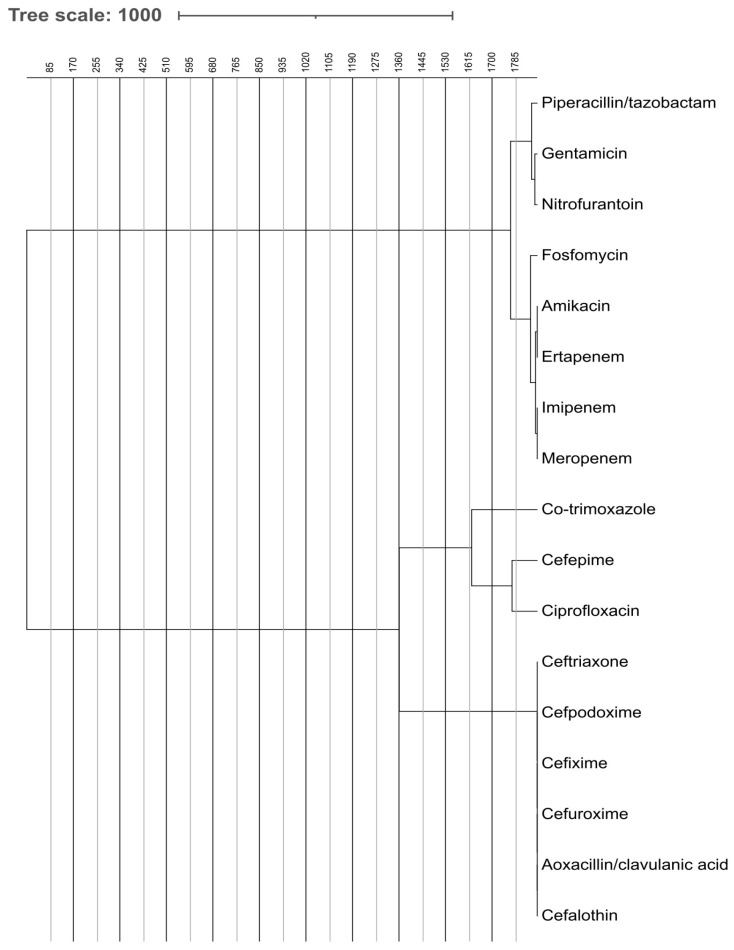
Comparative Analysis of Antibiotic Effectiveness Using Dendrogram for ESBL-Producing Bacteria. The figure shows dendrogram based comparative analysis illustrating the clustering of ESBL-producing bacteria according to their antibiotic effectiveness and resistance profiles.

**Figure 7 pathogens-14-01254-f007:**
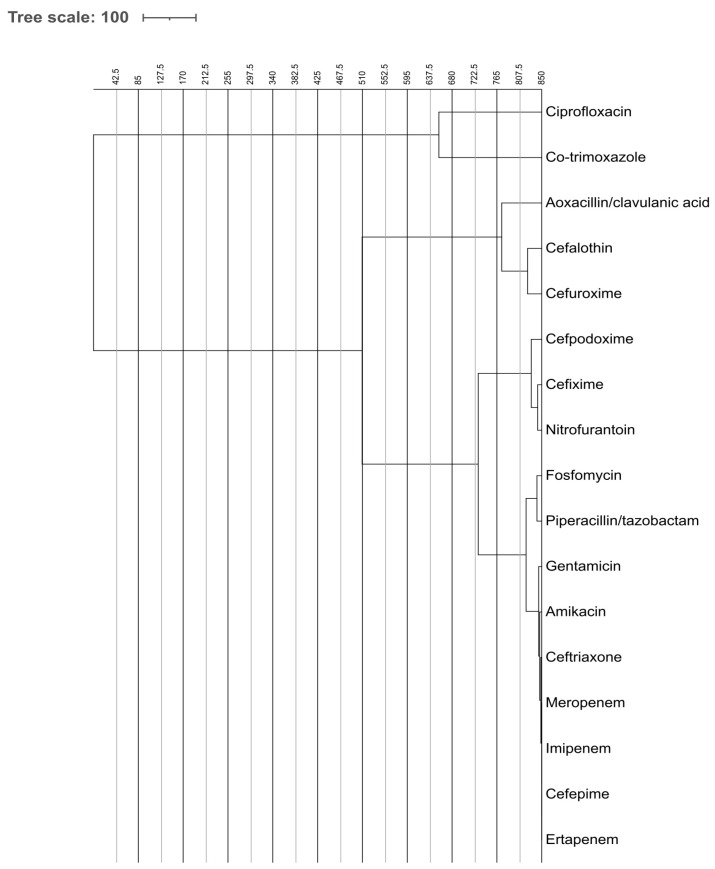
Comparative Analysis of Antibiotic Effectiveness Using Dendrogram for non-ESBL-Producing Bacteria. The figure shows dendrogram based comparative analysis illustrating the clustering of non-ESBL-producing bacteria according to their antibiotic effectiveness and resistance profiles.

**Table 1 pathogens-14-01254-t001:** Demographic Distribution of Positive Bacterial Cultures. Count of bacterial isolates by gender and age group, with total and percentage contribution of each organism to all positive cultures.

Result	Gender		Age Group (Years)								Total n (%)
	Male	Female	<1	2–5	5–15	16–30	31–45	46–60	61–75	>75	
**Positive Cultures**	441	2188	66	89	169	398	834	353	408	312	2629
**Organisms**											
**Enterobacterales**											
*Citrobacter freundii*	1	3	0	1	0	0	1	0	1	1	4 (<1)
*Citrobacter koseri*	7	35	1	0	1	6	13	10	8	3	42 (1.6)
*Citrobacter sedlakii*	0	1	0	0	0	0	0	0	1	0	1 (<1)
*Citrobacter species*	2	3	0	3	0	0	2	0	0	0	5 (<1)
*Escherichia coli*	196	1536	34	62	126	292	611	239	241	127	1732 (65.9)
*Enterobacter cloacae*	13	23	3	0	2	5	6	5	7	8	36 (1.4)
*Klebsiella pneumoniae*	77	365	12	2	19	66	132	61	91	59	442 (16.8)
*Klebsiella aerogenes*	7	14	1	0	2	5	8	1	4	0	21 (<1)
*Klebsiella oxytoca*	2	3	0	0	0	0	0	2	2	1	5 (<1)
*Morganella morganii*	2	6	0	3	1	0	0	1	1	2	8 (<1)
*Proteus mirabilis*	14	19	0	7	3	1	0	3	5	14	33 (1.3)
*Providencia species*	0	1	0	0	0	0	0	0	0	1	1 (<1)
*Serratia marcescens*	5	3	0	0	2	0	1	1	1	3	8 (<1)
*Serratia fonticola*	0	1	0	0	0	0	1	0	0	0	1 (<1)
*Salmonella species*	0	1	0	1	0	0	0	0	0	0	1 (<1)
** *Enterococcaceae* **											
*Enterococcus faecalis*	44	64	6	2	7	11	30	10	17	25	108 (4.1)
*Enterococcus faecium*	11	9	2	0	0	1	0	1	6	10	20 (<1)
*Enterococcus avium*	1	0	0	0	0	0	0	0	0	1	1 (<1)
** *Staphylococcaceae* **											
*Staphylococcus aureus*	11	8	0	0	0	4	5	2	3	5	19 (<1)
*Staphylococcus epidermidis*	0	1	0	0	0	0	0	0	1	0	1 (<1)
*Staphylococcus lugdunensis*	1	2	0	0	0	0	2	0	0	1	3 (<1)
*Staphylococcus saprophyticus*	1	18	0	1	1	3	11	3	0	0	19 (<1)
** *Pseudomonadaceae* **											
*Pseudomonas aeruginosa*	43	65	7	7	5	3	8	14	17	47	108 (4.1)
**Moraxellaceae**											
*Acinetobacter baumannii*	3	7	0	0	0	1	3	0	2	4	10 (<1)

**Table 2 pathogens-14-01254-t002:** Antibiotic Resistance Patterns in Non-ESBL and ESBL-Producing *E. coli.* The table summarizes resistance rates to commonly tested antibiotics among non-ESBL and ESBL isolates of *E. coli*, including overall resistance percentages and statistical significance.

Antibiotic	Non-ESBL Resistant	ESBL Resistant	Total Resistant n (%)	*p*-Value
**Amoxicillin/clavulanic acid**	157	530	687 (39.7)	*p* < 0.001
**Cephalothin**	120	530	650 (37.5)	*p* < 0.001
**Cefuroxime**	101	530	631 (36.4)	*p* < 0.001
**Cefixime**	53	530	583 (33.7)	*p* < 0.001
**Cefpodoxime**	38	530	568 (32.8)	*p* < 0.001
**Ceftriaxone**	4	530	534 (30.8)	*p* < 0.001
**Cefepime**	3	368	371 (21.4)	*p* < 0.001
**Ciprofloxacin**	427	433	860 (49.7)	*p* < 0.001
**Amikacin**	1	3	4 (0.2)	0.165
**Gentamicin**	6	36	42 (2.4)	*p* < 0.001
**Co-trimoxazole**	289	253	542 (31.3)	*p* < 0.001
**Fosfomycin**	12	15	27 (1.6)	*p* < 0.010
**Nitrofurantoin**	47	29	76 (4.4)	0.152
**Piperacillin/tazobactam**	18	41	59 (3.4)	*p* < 0.001
**Ertapenem**	3	3	6 (0.3)	0.556
**Imipenem**	3	0	3 (0.2)	0.600
**Meropenem**	3	0	3 (0.2)	0.600

**Table 3 pathogens-14-01254-t003:** Antibiotic Resistance Patterns in Non-ESBL and ESBL-Producing *Klebsiella species.* The table summarizes resistance rates to commonly tested antibiotics among non-ESBL and ESBL isolates of *Klebsiella species*, including overall resistance percentages and statistical significance.

	Non-ESBL Resistant	ESBL Resistant	Total Resistant n (%)	*p*-Value
**Amoxicillin/clavulanic acid**	68	97	165 (35.3)	*p* < 0.001
**Cephalothin**	61	97	158 (33.8)	*p* < 0.001
**Cefuroxime**	48	97	145 (31)	*p* < 0.001
**Cefixime**	24	97	121 (25.9)	*p* < 0.001
**Cefpodoxime**	20	97	117 (25)	*p* < 0.001
**Ceftriaxone**	12	97	109 (23.3)	*p* < 0.001
**Cefepime**	12	70	82 (17.5)	*p* < 0.001
**Ciprofloxacin**	51	80	131 (28)	*p* < 0.001
**Amikacin**	8	1	9 (1.9)	0.759
**Gentamicin**	9	1	10 (2.1)	0.651
**Co-trimoxazole**	35	69	104 (22.2)	*p* < 0.001
**Nitrofurantoin**	137	40	177 (37.8)	0.072
**Piperacillin/tazobactam**	23	14	37 (7.9)	*p* < 0.050
**Ertapenem**	12	2	14 (3)	0.788
**Imipenem**	12	3	15 (3.2)	*p* = 1
**Meropenem**	12	1	13 (2.8)	0.407
**Tigecycline**	2	0	2 (0.4)	*p* = 1

**Table 4 pathogens-14-01254-t004:** Distribution of VRE, MRSA, and CRE in Different Patient Settings. The table illustrates the occurrence of major multidrug resistant organisms stratified by inpatient and outpatient. VRE: vancomycin resistant *Enterococcus*, MRSA: methicillin resistant *Staphylococcus aureus*, CRE: carbapenem resistant Enterobacterales.

	Status	Inpatient	Outpatient
**VRE**	Negative	24 (20%)	96 (80%)
	Positive	7 (77.8%)	2 (22.2%)
**MRSA**	Negative	3 (20%)	12 (80%)
	Positive	1 (25%)	3 (75%)
**CRE**	Negative	120 (5.2%)	2205 (94.8%)
	Positive	6 (40%)	9 (60%)

## Data Availability

The original contributions presented in this study are included in the article. Further inquiries can be directed to the corresponding author(s).

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
