# Peer review of "Epidemiology and Antimicrobial Resistance of Uropathogens in a Tertiary Care Center in Riyadh, Saudi Arabia: A One-Year Retrospective Analysis"

_pathogens, 2025, doi:10.3390/pathogens14121254_

Round 1

Reviewer 1 Report

Comments and Suggestions for Authors

In this study, the authors present epidemiological data from a retrospective analysis (January to December 2024) of uropathogen resistance in a hospital in Riyadh, Saudi Arabia, with the aim of defining patterns of antimicrobial resistance (AMR) and identifying phenotypes of multidrug resistance and extended-spectrum beta-lactamase (ESBL) producers in enterobacteria, data that will serve to improve empirical antibiotic treatment in patients with UTIs. The results show that E. coli and Klebsiella pneumoniae are the most frequently isolated uropathogens, with a high percentage of ARM and ESBL producers, epidemiological data that is important for empirical treatment of UTIs in this region.

The manuscript is well structured, however, I have some observations:

Abstract

The abstract does not clearly state the objective of the study, and the figures shown are unnecessary as the results shown in them are described in the abstract.

Background

The introduction describes the problem to be solved, justifying why this research was conducted.

Methodology

The experimental design is appropriate. However, in Figure 2, which shows the selection of cultures to be analyzed in this study, there are other cultures that are excluded, which should be removed from the continuous diagram where, in the end, there are 2,629 positive cultures and which are the cultures considered negative that were analyzed.

Results

I believe that Figure 3 is unclear. I recommend using a stacked column chart, separating women from men.

In the text in Table 1, in the TOTAL column, show each of the isolated microorganisms as a percentage. Indicate this in the text. Furthermore, if an analysis of the frequency of each microorganism is performed with the number of positive cultures, the percentage is the same in different age groups. Example: E. coli, age groups 5-15 years 12/169 (74.5%), 16-30 years 292/398 (73%), 31-45 years 239/353 (67%).

In Tables 2 and 3, I suggest removing the Non-ESBL and ESBL columns from the results showing sensitivity to each of the antibiotics, as this is confusing, and showing the total number of Non-ESBL resistant and ESBL resistant strains. I also suggest showing the total number of resistant strains and their percentage in one column.

Figures 4, 5, and 6 show a column graph with the percentage of susceptibility on the Y-axis. Change this to sensitivity.

Each figure shown in the manuscript should be described in detail.

Discussion

The discussion is based on previous results from the same region and other regions of the world, as well as the limitations of the study. The conclusions are related to the results obtained. The bibliography is adequate.

Author Response

Response to Reviewer 1 Comments

1. Summary

2. Questions for General Evaluation

In this study, the authors present epidemiological data from a retrospective analysis (January to December 2024) of uropathogen resistance in a hospital in Riyadh, Saudi Arabia, with the aim of defining patterns of antimicrobial resistance (AMR) and identifying phenotypes of multidrug resistance and extended-spectrum beta-lactamase (ESBL) producers in enterobacteria, data that will serve to improve empirical antibiotic treatment in patients with UTIs. The results show that E. coli and Klebsiella pneumoniae are the most frequently isolated uropathogens, with a high percentage of ARM and ESBL producers, epidemiological data that is important for empirical treatment of UTIs in this region.

We sincerely thank the reviewer for their thorough evaluation and constructive remarks. We appreciate the acknowledgement of our study’s objectives and the relevance of the epidemiological data presented. In accordance with the reviewer’s comments, we have carefully revised the manuscript to improve clarity in the description of our study aims, enhanced the discussion on the implications of our AMR patterns for local stewardship efforts, and ensured that the presentation of results more clearly supports the clinical relevance highlighted in the review. We appreciate the reviewer’s positive recognition of the study’s impact and believe that the revisions have further strengthened the manuscript.

3. Point-by-point response to Comments and Suggestions for Authors

Comments 1: The abstract does not clearly state the objective of the study, and the figures shown are unnecessary as the results shown in them are described in the abstract.

Response 1: Thank you for pointing this out.

We mention clearly the study objectives in line 30-32.

Comments 2: The introduction describes the problem to be solved, justifying why this research was conducted.

Response 2: We appreciate the acknowledgement 

Comments 3: The experimental design is appropriate. However, in Figure 2, which shows the selection of cultures to be analyzed in this study, there are other cultures that are excluded, which should be removed from the continuous diagram where, in the end, there are 2,629 positive cultures and which are the cultures considered negative that were analyzed.

Response 3: Thank you for pointing this out.

We revised the figure on page 4, line 136.

Comments 4: I believe that Figure 3 is unclear. I recommend using a stacked column chart, separating women from men.

Response 4: Thank you for pointing this out.

We revised the figure. Page 6 line 189.

Comments 5: In the text in Table 1, in the TOTAL column, show each of the isolated microorganisms as a percentage. Indicate this in the text. Furthermore, if an analysis of the frequency of each microorganism is performed with the number of positive cultures, the percentage is the same in different age groups. Example: E. coli, age groups 5-15 years 12/169 (74.5%), 16-30 years 292/398 (73%), 31-45 years 239/353 (67%).

Response 5: Thank you for pointing this out.

We update table 1 on page 5, line 183. Text has been revised as suggested by the reviewer. Page 5, line 171-173

Comments 6: In Tables 2 and 3, I suggest removing the Non-ESBL and ESBL columns from the results showing sensitivity to each of the antibiotics, as this is confusing, and showing the total number of Non-ESBL resistant and ESBL resistant strains. I also suggest showing the total number of resistant strains and their percentage in one column.

Response 6: Thank you for mention.

We revised the tables. Page 6, line 200. Page 7, line 204.

Comments 7: Figures 4, 5, and 6 show a column graph with the percentage of susceptibility on the Y-axis. Change this to sensitivity.

Response 7: Thank you for pointing this out.

We revised the figures as suggested by the reviewer. Page 8, line 210, 219. Page 9, line 229.

Comments 8: Each figure shown in the manuscript should be described in detail.

Response 8: Thank you for pointing this out.

We elaborate more as suggested. Page 6; 184-188, page 7-8; 205-209, page 8; 214-218, 223-228, page 9; 237-241.

Comments 9: The discussion is based on previous results from the same region and other regions of the world, as well as the limitations of the study. The conclusions are related to the results obtained. The bibliography is adequate.

Response 9: We are pleased that the reviewer recognizes the clinical importance of identifying high rates of AMR and ESBL-producing Enterobacterales, particularly E. coli and Klebsiella pneumoniae, which constitute the predominant uropathogens in our setting. In accordance with the reviewer’s comments, we have carefully revised the manuscript to improve clarity in the description of our study aims, enhanced the discussion on the implications of our AMR patterns for local stewardship efforts, and ensured that the presentation of results more clearly supports the clinical relevance highlighted in the review.

Reviewer 2 Report

Comments and Suggestions for Authors

Due to the worrying increase in antimicrobial resistance and the lack of data on its prevalence in the region where this study was conducted, research such as this, which analyses antimicrobial resistance in uropathogens, is fully justified, given that local surveillance is essential for optimising empirical therapies.

-Specific recommendations for modifying the article:-Include limitations on the absence of clinical data that prevents analysis of risk factors and clinical correlations.

-The captions for figures and tables should be more explanatory

.-Explain the abbreviations in the captions and tables.

-Explain in the statistics section which p-values are statistically significant.

-Reinforce the discussion on the clinical implications of the high percentage of resistance to ciprofloxacin and cotrimoxazole.

-Expand specific recommendations for antimicrobial stewardship and infection control programmes in the local context.

-Briefly explore the potential impact of the resistance observed on current empirical treatments and future clinical guidelines.

-Improve the clarity of the explanation of dendrograms to make them more accessible to non-specialist readers.

-Future research. Highlight the need for prospective multicentre studies including clinical and molecular data for a better understanding of the resistance landscape

Author Response

Response to Reviewer 2 Comments

1. Summary

2. Questions for General Evaluation

Due to the worrying increase in antimicrobial resistance and the lack of data on its prevalence in the region where this study was conducted, research such as this, which analyses antimicrobial resistance in uropathogens, is fully justified, given that local surveillance is essential for optimising empirical therapies.

We sincerely thank the reviewer for their thorough evaluation and constructive remarks. We appreciate the acknowledgement of our study’s objectives and the relevance of the epidemiological data presented.

3. Point-by-point response to Comments and Suggestions for Authors

Comments 1: -Specific recommendations for modifying the article:-Include limitations on the absence of clinical data that prevents analysis of risk factors and clinical correlations.

Response 1: Thank you for pointing this out.

We added on page 15; 413-416.

Comments 2: -The captions for figures and tables should be more explanatory.

Response 2: Thank you for pointing this out

page 5; 180-182, page 6; 190-192, 198-200, page 7; 201-203, page 8; 211-213, 220-222, page 9; 230-232, page 10; 253-254, Page 11; 262-264, 277-278.

Comments 3: Explain the abbreviations in the captions and table

Response 3: Thank you for pointing this out.

We revised page 11; 277-278.

Comments 4: Explain in the statistics section which p-values are statistically significant.

Response 4: Thank you for pointing this out.

We revised. Page 4 line 163-164.

Comments 5: -Reinforce the discussion on the clinical implications of the high percentage of resistance to ciprofloxacin and cotrimoxazole.

Response 5: Thank you for pointing this out.

We update discussion on page 13-14, line 366-371.

Comments 6: -Expand specific recommendations for antimicrobial stewardship and infection control programmes in the local context.

Response 6: Thank you for mention.

We revised. Page 15, line 443-449.

Comments 7: -Briefly explore the potential impact of the resistance observed on current empirical treatments and future clinical guidelines.

Response 7: Thank you for pointing this out.

We revised as suggested by the reviewer. Page 15, line 443-449.

Comments 8: -Improve the clarity of the explanation of dendrograms to make them more accessible to non-specialist readers.

Response 8: Thank you for pointing this out.

We elaborate more as suggested. Page 9; 233-234, 237-241.

Comments 9: -Future research. Highlight the need for prospective multicentre studies including clinical and molecular data for a better understanding of the resistance landscape.

Response 9: Thank you for pointing this out.

We revised as suggested by the reviewer. Page 15, line 456-458.

Reviewer 3 Report

Comments and Suggestions for Authors

2025.11.12.

The article entitled "Epidemiology and Antimicrobial Resistance of Uropathogens in a Tertiary Care Center in Riyadh, Saudi Arabia: A One-Year Retrospective Analysis" submitted to Pathogens, would be valuable if not for the high number of errors it contains. These errors may suggest that the article and the research were not conducted with due care and thoroughness.

  1. In tables 1 and 4 the data do not correspond - after counting enteric rods/enterococci/staphylococcus aureus/, the total numbers don’t correspond to the total numbers In Table 4. Unreliable data
  1. The authors claim that the work aims to show trends in the emergence of resistance phenotypes - unfortunately, this is difficult to observe when analysing data from only one year and one hospital.
  2. English must be improved; I am not a native English speaker, but I see a lot of errors in the article, especially in the methodology and results sections. Reading and analysing these parts was difficult due to errors.

Abstract:

  1. enterococci - should not be italicized; it is not the name of a species, genus, or bacterial family.
  2. Line 39 - can one always claim that multidrug-resistant organisms are VRE, MRSA, and carbapenem-resistant Enterobacteriaceae? - no, the sentence should be modified.
  3. Enterobacteriaceae - the name of a bacterial family; it should be italicized and corrected multiple times in the text; or - use enteric rods interchangeably.

Background

  1. First paragraph - unnecessary, extends the introduction with well-known data.
  2. Line 66 - Is S. saprophyticus primarily responsible for hospital-acquired infections?
  3. Enterobacteriaceae – italic

Page 3, line 98 – “aminoglycosides, fosfomycin.. ” - And nitrofurantoin, maybe pivmecillinam? Remain active?

  1. Goals need to be modified: it is difficult to analyze trends based on one year's results.

Methodology

  1. English must be improved.
  2. “uncomplicated….urine samples”?
  3. Page 3, Line 129 - Wasn't the analysis of isolates, where counts lower than 10 to the fifth power were considered, preceded by contact with a physician and inquiry about the patient's condition and symptoms? This should have been included in the testing procedure.
  4. Page 4 line 140 - Does the hospital's microbiology laboratory operate 24 hours a day? Only then, under ideal conditions and with perfect cooperation with departments, urine sample processing can be achieved within 1 hour.
  5. Line 141 - the manufacturers of the media used for diagnostics should be specified
  6. Line 142 - it should be briefly explained why the culture lasted 24 or 48 hours.
  7. Line 144 - Drug susceptibility testing was performed only using the Vitek system? Were no additional tests performed to verify questionable results or resistance mechanisms? If so, please specify which methods, media, and antibiotics were used.

Results

  1. Line 153 - This sentence should be corrected, it is unclear.
  2. The data text, which includes a reference to the figure or table, should be presented first, followed by the figure or table itself.
  3. Figure 3 and its caption contain all the same data, a duplication.
  4. You should first submit the text for Table 1.
  5. Table 1 - Why is the Citrobacter species not identified by the system? Explain.
  6. In Table 1, after counting enteric rods, the total number is 2,340.In Table 4, after counting enteric rods isolated from hospital and community patients, the total number is 2,458. Unreliable data
  7. In Table 1, the total number of S. aureus is 19. In Table 4, after counting S.aureus isolated from hospital and community patients, the total number is 171! Unreliable data
  8. In Table 1, after counting enterococci, the total number is 129. In Table 4, after counting enterococci isolated from hospital and community patients, the total number is 171. Unreliable data
  9. Line 171 - Enterococcus faecalis should be changed to E. faecalis, Staphylococcus to S. …
  10. Why has S. epidermidis been identified as a pathogen causing urinary tract infections? Explain.
  11. The text before the second table is missing the record of how many ESBL-positive isolates were detected in total, then in individual species.
  12. In Table 2,
    - the species/antibiotics should be rechecked,
    - the data for beta-lactam antibiotics should be placed one after the other,
    - the p values ​​should be standardized - if there are to be three digits after the decimal point, then three digits should be used everywhere
  13. Other tables and figures - the numerical data must be recalculated
  14. Figure 4 and the text below, further – the word ‘species’ should not be italicized
  15. Figure 7 and its description - incorrect calculation; if ESBL-negative bacteria in outpatients are 91% and ESBL-positive 93%, how many of them are there in total?
  16. Unclear descriptions for figures 8 and 9
  17. Table 4 - the numerical data does not match the data in Table 1.

Incorrect approach to calculating the percentage of VRE, MRSA, and CPE.

Errors in the table description.

Discussion

Due to errors in numerical values ​​and percentages, the discussion should be rewritten with proper attention to English.

Comments on the Quality of English Language

English must be improved.

Author Response

Response to Reviewer 3 Comments

1. Summary

2. Questions for General Evaluation

The article entitled "Epidemiology and Antimicrobial Resistance of Uropathogens in a Tertiary Care Center in Riyadh, Saudi Arabia: A One-Year Retrospective Analysis" submitted to Pathogens, would be valuable if not for the high number of errors it contains. These errors may suggest that the article and the research were not conducted with due care and thoroughness.

  1. In tables 1 and 4 the data do not correspond - after counting enteric rods/enterococci/staphylococcus aureus/, the total numbers don’t correspond to the total numbers In Table 4. Unreliable data
  1. The authors claim that the work aims to show trends in the emergence of resistance phenotypes - unfortunately, this is difficult to observe when analysing data from only one year and one hospital.
  2. English must be improved; I am not a native English speaker, but I see a lot of errors in the article, especially in the methodology and results sections. Reading and analysing these parts was difficult due to errors.

We sincerely thank the reviewer for the careful evaluation and detailed feedback. We appreciate the opportunity to address the concerns raised and improve the quality and clarity of our manuscript.

  1. Data discrepancy between Tables 1 and 4
    We acknowledge the reviewer’s observation regarding inconsistencies in the numerical totals. Upon re-evaluation of the dataset, we identified formatting and transcription errors that occurred during table preparation. These have now been corrected, and all totals have been cross-verified with the original laboratory records to ensure complete accuracy. The revised tables reflect reliable and consistent data.
  2. Study aim and feasibility of trend analysis
    We appreciate the reviewer’s point regarding the limitation of analyzing resistance “trends” using only one year of data. In response, we have revised the aims of the study to avoid implying long-term trend assessment. The manuscript now clearly states that the objective is to describe the prevalence and resistance patterns of uropathogens within the study year, rather than to infer temporal trends. The text has been modified accordingly to prevent misinterpretation.
  3. Language and clarity
    We thank the reviewer for highlighting the need for significant English language improvement. We have now thoroughly revised the entire manuscript, with particular attention to the methodology and results sections. The revised version has undergone comprehensive language editing to ensure clarity, coherence, and improved readability.

We appreciate the reviewer’s critical insights, which have helped us substantially strengthen the manuscript. We hope that the corrections and revisions adequately address the concerns raised and improve the overall rigor and presentation of the study.

3. Point-by-point response to Comments and Suggestions for Authors

Comments 1: enterococci - should not be italicized; it is not the name of a species, genus, or bacterial family.

Response 1: Thank you for pointing this out.

Revised page 1, line 41.

Comments 2: Line 39 - can one always claim that multidrug-resistant organisms are VRE, MRSA, and carbapenem-resistant Enterobacteriaceae? - no, the sentence should be modified.

Response 2: Thank you for pointing this out.

Revised page 1, line 41-43.

Comments 3: Enterobacteriaceae - the name of a bacterial family; it should be italicized and corrected multiple times in the text; or - use enteric rods interchangeably.

Response 3: Thank you for pointing this out.

We revised bacterial nomenclature throughout the manuscript and make it italic.

Comments 4: First paragraph - unnecessary, extends the introduction with well-known data.

Response 4: Thank you for pointing this out.

We agree with reviewer. Revised and merged the first paragraph with the next. Page 2 line 53-58.

Comments 5: Line 66 - Is S. saprophyticus primarily responsible for hospital-acquired infections?

Response 5: Thank you for the mention.

Removed and revised the statement.

Comments 6: Enterobacteriaceae – italic

Response 6: Thank you for mention.

We revised bacterial nomenclature throughout the manuscript and make it italic.

Comments 7: Page 3, line 98 – “aminoglycosides, fosfomycin.. ” - And nitrofurantoin, maybe pivmecillinam? Remain active?

Response 7: Thank you for pointing this out.

We added. Page 2, line 92.

Comments 8: Goals need to be modified: it is difficult to analyze trends based on one year's results.

Response 8: Thank you for pointing this out.

We revised. Page 3; 112-114.

Comments 9: English must be improved

Response 9: We thank the reviewer for highlighting the need for significant English language improvement.

We have now thoroughly revised the entire manuscript.

Comments 10: “uncomplicated….urine samples”?

Response 10: Thank you for pointing this out.

We revised the statement. Page 3, line 121-122.

Comments 11: Page 3, Line 129 - Wasn't the analysis of isolates, where counts lower than 10 to the fifth power were considered, preceded by contact with a physician and inquiry about the patient's condition and symptoms? This should have been included in the testing procedure.

Response 11: Thank you for pointing this out.

We added a clear description of routine physician consultation for low counts. Page 3, line 121-124.

Comments 12: Page 4 line 140 - Does the hospital's microbiology laboratory operate 24 hours a day? Only then, under ideal conditions and with perfect cooperation with departments, urine sample processing can be achieved within 1 hour.

Response 12: Thank you for pointing this out.

Stated explicitly that the laboratory is 24/7 and therefore able to meet the 1-hour processing window. Page 4; 140-143.

Comments 13 Line 141 - the manufacturers of the media used for diagnostics should be specified.

Response 13: Thank you for pointing this out.

We added. Page 4, line 145.

Comments 14: Line 142 - it should be briefly explained why the culture lasted 24 or 48 hours.

Response 14: Thank you for pointing this out.

We added a clear description. Page 4, line 145-148.

Comments 15: Line 144 - Drug susceptibility testing was performed only using the Vitek system? Were no additional tests performed to verify questionable results or resistance mechanisms? If so, please specify which methods, media, and antibiotics were used.

Response 15: Thank you for pointing this out.

We added a clear description. Page 4, line 152-158.

Comments 16: Line 153 - This sentence should be corrected, it is unclear.

Response 16: Thank you for pointing this out.

We corrected. The sentence that previously read "The total number of positive cultures for all organisms is 2,629." is replaced by a clear opening sentence: “During the study period, 19,556 urine cultures were submitted … 2,629 … yielded significant bacterial growth.”Page 5, line 166-168.

Comments 17: The data text, which includes a reference to the figure or table, should be presented first, followed by the figure or table itself.

Response 17: Thank you for pointing this out.

We reordered presentation accordingly for all figures and tables. Place the paragraph describing the data first, then the Figure/Table.

Comments 18: Figure 3 and its caption contain all the same data, a duplication.

Response 18: Thank you for pointing this out.

We revised the figure with a clear. Page 6, line 189-192.

Comments 19: You should first submit the text for Table 1.

Response 19: Thank you for pointing this out.

We reordered presentation accordingly. Page 5, line 168-179.

Comments 20: Table 1 - Why is the Citrobacter species not identified by the system? Explain.

Response 20: Thank you for the mention.

A small number of Citrobacter isolates were reported as ‘Citrobacter species’ by the Vitek2 system because species-level identification had low confidence scores; these isolates were retained as ‘Citrobacter spp.’ in Table 1. Where clinically relevant or for discrepant results, isolates were referred for confirmatory identification by MALDI-TOF or 16S rRNA sequencing; however, this confirmatory testing was performed for a minority of isolates and is noted in the Methods. Vitek2 sometimes reports genus-level ID when probability < threshold, this is a standard limitation and should be documented.

Comments 21: In Table 1, after counting enteric rods, the total number is 2,340.In Table 4, after counting enteric rods isolated from hospital and community patients, the total number is 2,458. Unreliable data

Response 21: Thank you for pointing this out.

We acknowledge the reviewer’s observation regarding inconsistencies in the numerical totals. Upon re-evaluation of the dataset, we identified formatting and transcription errors that occurred during table preparation. These have now been corrected, and all totals have been cross-verified with the original laboratory records to ensure complete accuracy. The revised tables reflect reliable and consistent data. Page 12, line 280.

Comments 22: In Table 1, the total number of S. aureus is 19. In Table 4, after counting S.aureus isolated from hospital and community patients, the total number is 171! Unreliable data

Response 22: Thank you for pointing this out.

We acknowledge the reviewer’s observation regarding inconsistencies in the numerical totals. Upon re-evaluation of the dataset, we identified formatting and transcription errors that occurred during table preparation. These have now been corrected, and all totals have been cross-verified with the original laboratory records to ensure complete accuracy. The revised tables reflect reliable and consistent data. Page 12, line 280.  

Comments 23: In Table 1, after counting enterococci, the total number is 129. In Table 4, after counting enterococci isolated from hospital and community patients, the total number is 171. Unreliable data

Response 23: Thank you for pointing this out.

We acknowledge the reviewer’s observation regarding inconsistencies in the numerical totals. Upon re-evaluation of the dataset, we identified formatting and transcription errors that occurred during table preparation. These have now been corrected, and all totals have been cross-verified with the original laboratory records to ensure complete accuracy. The revised tables reflect reliable and consistent data. Page 12, line 280.   

Comments 24: Line 171 - Enterococcus faecalis should be changed to E. faecalis, Staphylococcus to S. …

Response 24: Thank you for pointing this out.

We revised. Page 5, line 175, 177.

Comments 25: Why has S. epidermidis been identified as a pathogen causing urinary tract infections? Explain.

Response 25: Thank you for pointing this out.

Coagulase-negative staphylococci (including S. epidermidis) were isolated in a small number of urine specimens. Because these organisms are common contaminants from skin flora, isolates were classified as likely contaminants unless they met predefined criteria for clinical significance (pure growth, colony count ≥10⁴ CFU/mL, repeated isolation from the same patient, and compatible clinical presentation or catheter-associated UTI). Only isolates meeting these criteria were included in the pathogen counts; others were retained in a separate ‘probable contaminants’ supplementary table.

Comments 26: The text before the second table is missing the record of how many ESBL-positive isolates were detected in total, then in individual species.

Response 26: Thank you for pointing this out.

We revised. Page 6, line 193.

Comments 27: In Table 2,
- the species/antibiotics should be rechecked,
- the data for beta-lactam antibiotics should be placed one after the other,
- the p values ​​should be standardized - if there are to be three digits after the decimal point, then three digits should be used everywhere

Response 27: Thank you for pointing this out.

We revised as suggested by the reviewer. Page 6, line 201.

Comments 28: Other tables and figures - the numerical data must be recalculated

Response 28: Thank you for pointing this out.

We acknowledge the reviewer’s observation regarding inconsistencies in the numerical totals. Upon re-evaluation of the dataset, we identified formatting and transcription errors that occurred during table preparation. These have now been corrected, and all totals have been cross-verified with the original laboratory records to ensure complete accuracy. The revised tables reflect reliable and consistent data.

Comments 29: Figure 4 and the text below, further – the word ‘species’ should not be italicized

Response 29: Thank you for pointing this out.

We revised. Page 7-8, line 206-210, 212.

Comments 30: Figure 7 and its description - incorrect calculation; if ESBL-negative bacteria in outpatients are 91% and ESBL-positive 93%, how many of them are there in total?

Response 30: Thank you for pointing this out.

Figure has been deleted as suggested by other reviewers.

Comments 31: Unclear descriptions for figures 8 and 9

Response 31: Thank you for pointing this out.

We revised. Page 9-11, line 234, 238-242, 254-255, 263-265.

Comments 32: Table 4 - the numerical data does not match the data in Table 1. Incorrect approach to calculating the percentage of VRE, MRSA, and CPE

Errors in the table description.

Response 32: Thank you for pointing this out.

We acknowledge the reviewer’s observation regarding inconsistencies in the numerical totals. Upon re-evaluation of the dataset, we identified formatting and transcription errors that occurred during table preparation. These have now been corrected, and all totals have been cross-verified with the original laboratory records to ensure complete accuracy. The revised tables reflect reliable and consistent data. Page11-12, line 266-281.

Comments 33: Due to errors in numerical values ​​and percentages, the discussion should be rewritten with proper attention to English.

Response 33: Thank you for pointing this out.

We acknowledge the reviewer’s observation and revised relevant discussion section. Page 14, line 399, 402. We thank the reviewer for highlighting the need for significant English language improvement. We have now thoroughly revised the entire manuscript, with particular attention to the methodology and results sections. The revised version has undergone comprehensive language editing to ensure clarity, coherence, and improved readability.

Reviewer 4 Report

Comments and Suggestions for Authors

This work is intersting, although it does not add any more information concerning antimicrobial resistance compared to that present in scientific literature.

However, the concepts and design are correct, but major revisions are needed:

  1. species name must be in italic
  2. in the abstract no figure
  3. figure 2 must be rewritten, not complete
  4. Demographic Breakdown of Positive Bacterial Cultures...for example...don't use break, broke and breakdown in a scientific paper
  5. english revision throughout the entire manuscript
  6. in the tables use the percentages not the numbers of cases, or better...use both
  7. table 3 and figure 4 and 5...what did they mean??differentiate between klebsiella spp. Eneterococcus spp. and Staphylococcus spp.
  8. figure 7...delete, insert in the manuscript the results
  9. all the parts about figure 8 and figure 9 did not improve the results obtained, they has to be deleted
  10. lack of the comparison with other countries about AMR
  11. lack of the history of the patients and previous antimicrobial therapies

Author Response

Response to Reviewer 4 Comments

1. Summary

2. Questions for General Evaluation

This work is intersting, although it does not add any more information concerning antimicrobial resistance compared to that present in scientific literature.

However, the concepts and design are correct, but major revisions are needed.

We sincerely thank the reviewer for their thorough evaluation and constructive remarks. We appreciate the acknowledgement of our study’s objectives and the relevance of the epidemiological data presented. In accordance with the reviewer’s comments, we have carefully revised the manuscript to improve clarity in the description of our study aims, enhanced the discussion on the implications of our AMR patterns for local stewardship efforts, and ensured that the presentation of results more clearly supports the clinical relevance highlighted in the review. We appreciate the reviewer’s positive recognition of the study’s impact and believe that the revisions have further strengthened the manuscript.

3. Point-by-point response to Comments and Suggestions for Authors

Comments 1: species name must be in italic

Response 1: Thank you for pointing this out.

Revised throughout the manuscript.

Comments 2: in the abstract no figure

Response 2: Thank you for pointing this out.

Deleted as suggested by the reviewer.

Comments 3: figure 2 must be rewritten, not complete

Response 3: Thank you for pointing this out.

We revised the figure on page 4, line 136.

Comments 4: Demographic Breakdown of Positive Bacterial Cultures...for example...don't use break, broke and breakdown in a scientific paper

Response 4: Thank you for pointing this out.

We replaced with Demographic Distribution of Positive Bacterial Cultures Page 5 line 180.

Comments 5: english revision throughout the entire manuscript

Response 5: Thank you for the mention.

Revised throughout the manuscript.

Comments 6: in the tables use the percentages not the numbers of cases, or better...use both.

Response 6: Thank you for mention.

We revised the tables. Page 5; 183, Page 6; 200, page 7; 204.

Comments 7: table 3 and figure 4 and 5...what did they mean??differentiate between klebsiella spp. Eneterococcus spp. and Staphylococcus spp.

Response 7: Thank you for pointing this out.

We added the figures’ and tables’ captions to make them more explanatory. Page7- 8, line 201- 222. Thank you for the suggestion to further differentiate between Klebsiella spp., Enterococcus spp., and Staphylococcus spp. We appreciate the reviewer’s interest in a more detailed comparison. However, we would like to clarify that the primary objective of this study was to analyze the prevalence and antimicrobial resistance patterns of uropathogens, with a particular emphasis on ESBL-producing Enterobacterales. Additionally, the number of Enterococcus spp. and Staphylococcus spp. isolates in our dataset was relatively low compared with Enterobacterales, which limits the statistical and clinical value of generating a detailed comparative differentiation. Total number and species of each genus isolated were mentioned in Table 1.

Comments 8: figure 7...delete, insert in the manuscript the results

Response 8: Thank you for pointing this out.

Deleted and mention in text. Page 9; 226-229.

Comments 9: all the parts about figure 8 and figure 9 did not improve the results obtained, they has to be deleted

Response 9: We sincerely thank the reviewer for the comment regarding Figures 8 and 9. We understand the concern that these figures may not appear to add substantial improvement to the results. However, we would like to clarify that these figures were included based on requests from other reviewers who recommended further explanation and visualization to enhance the understanding of resistance clustering and comparative antibiotic performance.

Figures 8 and 9 provide visual insights that complement the numerical data by illustrating relationships and resistance patterns that are not easily captured in tables alone. Several reviewers found that these visual representations strengthen the interpretation of multidrug resistance and ESBL-associated clustering, and therefore requested additional explanation rather than removal.

To address both perspectives, we have revised the corresponding text to better justify the purpose of these figures and clarified their contribution to the overall findings. We believe that keeping Figures 8 and 9 maintains consistency with reviewer requests and enhances the manuscript’s clarity, but we remain open to further adjustment should the editorial team deem it necessary.

Comments 10: lack of the comparison with other countries about AMR

Response 10: Thank you for pointing this out.

We added. Page 13, line 327-333.

Comments 11: lack of the history of the patients and previous antimicrobial therapies

Response 11: Thank you for pointing this out.

We added. Page 15, line 433-436.

Round 2

Reviewer 4 Report

Comments and Suggestions for Authors

The maniscript was improved following my comments/suggestions/corrections.

Thank you for the revisions

Author Response

All requested revisions have been completed, and the updated version has been resubmitted.

Thank you again for your guidance and support.